# Cross-Cultural Adaptation and Psychometric Properties of the Arabic Version of the Fall Risk Questionnaire

**DOI:** 10.3390/ijerph20085606

**Published:** 2023-04-21

**Authors:** Ahmad A. Alharbi, Hamad S. Al Amer, Abdulaziz A. Albalwi, Majed Y. Muthaffar, Yousef M. Alshehre, Hani F. Albalawi, Turky E. Alshaikhi

**Affiliations:** 1Department of Physical Therapy, Faculty of Applied Medical Sciences, University of Tabuk, Tabuk 71491, Saudi Arabia; 2Department of Physical Therapy, King Fahad Hospital, Madinah 42392, Saudi Arabia; 3Department of Languages and Translation, Faculty of Arts and Education, University of Tabuk, Tabuk 71491, Saudi Arabia

**Keywords:** falls, fall prevention, older adults, risk assessment

## Abstract

A simple, valid, and reliable self-assessment fall-risk questionnaire in Arabic is needed to significantly promote awareness and develop fall-prevention programs. This study translated and adapted the Fall Risk Questionnaire (FRQ) into Arabic and determined its validity and reliability among Saudi Arabian older adults in two phases: (1) cross-culturally adapting the FRQ into Arabic and (2) assessing the adapted questionnaire’s psychometric properties in two sessions with 110 Arabic-speaking participants aged ≥65 years. Pearson’s r showed that the Arabic FRQ had a significant moderate negative relationship with the Berg Balance Scale and fair-to-moderate positive correlations with Five Time Sit to Stand and Time Up and Go. The receiver operating characteristic curve indicated a significant area under the curve = 0.81. The cut-off score was 7.5 and associated with 73.7% sensitivity and 73.6% specificity. Internal consistency was estimated as good, with Cronbach’s α = 0.77. Deletion of item 1 slightly increased Cronbach’s α to 0.78. The Arabic FRQ demonstrated excellent test-retest reliability, with an intraclass correlation coefficient = 0.95 (95% CI: 0.92–0.97). It is highly valid and reliable in providing valuable data for evaluating fall risk in adults aged ≥65 years and for consulting a specialist for further investigation if necessary.

## 1. Background

Falls, defined as sudden events that result in a person “inadvertently coming to rest on the ground, floor or other lower level”, are a common problem affecting older adults (i.e., adults aged 65 and over) [1] (p. 1). Approximately one in four individuals aged 65 and over experiences at least one fall yearly [1]. The prevalence is even higher in Saudi Arabia. A recent study reported that more than half of the older adults living in Saudi Arabia had experienced at least one fall within the past year [2]. Factors that may explain the high prevalence of falling among older Saudi adults include low education level, history of previous fall, and having more than one chronic disease [2]. Falls usually lead to negative health consequences, including soft tissue injuries, fractures, loss of independence, fear of falling, depression, disability, reduced quality of life, and even death [3,4,5]. Therefore, fall prevention and fall-risk awareness are considered health priorities, especially when the high rate of falling among older adults is considered.

Early identification of individuals at increased risk for falls is essential for effective fall-prevention programs, and clinical guidelines recommend using fall-risk assessment tools to screen older adult populations for fall risk [6,7]. However, despite existing recommendations, many physicians and other healthcare providers find it impractical to incorporate fall-risk assessment tools into daily practice [8]. This is because most of fall-risk assessment tools are performance-based tests requiring both clinical space and time for administration [9]. For instance, these physical performance tests ask older adults to perform predefined activities such as walking along a 20-meter corridor for six minutes [10] or getting up from a chair and walking for three meters before returning to the chair [11].

In addition, older adults tend not to recognize that they might be personally at increased risk of falling, which may result in their delayed help-seeking [12]. Shankar et al. [12], 2017 found that fall-prevention behavior among older adults could be significantly influenced by fall-risk awareness. Therefore, identifying a simple and applicable fall-risk screening tool is very important for effective fall-prevention programs.

Rubenstein et al. [13] developed and validated the simple, brief, and fast 12-question self-rated Fall Risk Questionnaire (FRQ). It is included within leaflets prepared by the Centers for Disease Control and Prevention to provide older adults the ability to assess their own fall risk and to raise their awareness on seeking medical help as the next step [13]. Other countries have also recognized the importance of such a tool. Consequently, the FRQ was translated into the Turkish and Thai languages and showed high validity and reliability [14,15].

The availability of an easy self-rated FRQ could significantly contribute to raising awareness among older adults regarding risk of falling and thus seeking medical help when required. It can also be used in developing fall-prevention programs. Unfortunately, there is a lack of a simple, valid, and reliable self-assessment FRQ in Arabic. Twenty-two countries use Arabic as their official language, and around 422 million people speak Arabic around the world. Therefore, this study aimed to translate and adapt the FRQ into Arabic and to determine its validity and reliability among older adults in Saudi Arabia.

## 2. Methods

The present study was conducted in two main phases. In the first phase, we cross-culturally adapted the FRQ into Arabic. In the second phase, we assessed the adapted questionnaire’s psychometric properties, including its validity and reliability.

Using a convenient sample, the participants were older adults living in the community who were recruited voluntarily by personal communications, and the data were collected in the two outpatient physical therapy departments in Saudi Arabia. Individuals aged ≥65 years who could speak Arabic, walk independently, and follow the examiner’s instructions were included in the study. In contrast, individuals with neurological or orthopedic disorders that disturbed their walking and balance, uncontrolled hypertension, and cardiovascular disease were excluded.

The study protocol complied with the ethical approval criteria according to the rules and regulations of the National Committee of Bioethics, with approval from the Local Ethics Committee at the University of Tabuk (UT-195-45-2022). All participants signed an informed consent form prior to participation in the study.

### 2.1. Translation and Cross-Cultural Adaptation

Necessary permissions for the translation adaptation study were obtained from the authors who developed the questionnaire [13]. The FRQ’s translation and cross-cultural adaptation were conducted in five stages, as recommended by Beaton et al. [16].

#### 2.1.1. First Stage: Forward Translation

Two translators who are native Arabic speakers and proficient in English translated the questionnaire from English into Arabic. The first translator holds a Ph.D. degree in physical therapy from the United States with 15 years of experience and was aware of the FRQ concept. The second translator, who is a language expert, is an associate professor in the Languages and Translation Department at the University of Tabuk with neither a medical background nor awareness of the FRQ concept. Two versions of the translation were produced—referred to as “T1” and “T2”.

#### 2.1.2. Second Stage: Translation Synthesis

The authors arranged a meeting with the two translators to synthesize T1 and T2. They reached a consensus and formed a single questionnaire (T12).

#### 2.1.3. Third Stage: Backward Translation

Two translators who are native English speakers and proficient in the Arabic language translated the synthesized Arabic version of the FRQ (T12) back into English. Both translators teach English as a second language for Arab students in the Institute of Languages at the University of Tabuk. The translators had no medical background or access to the original version of the FRQ. In addition, they were asked not to search for the questionnaire. Two versions of the backtranslation were produced—referred to as “B1” and “B2”.

#### 2.1.4. Fourth Stage: Expert Committee

The expert committee included two of the authors, the translators, and a methodologist. The methodologist is a professor who holds a DrPH degree in Public Health and specialized in epidemiology and biostatistics. They discussed discrepancies or inconsistencies in the previous stages of the translation process by comparing the back-translated versions with the original FRQ. The expert committee made necessary changes to ensure clarity and suitability for the general Arabic public and provided the pre-final Arabic version of the FRQ for field testing.

#### 2.1.5. Fifth Stage: Content Validity and Pilot Testing

Twelve health professionals (four family physicians, four physical therapists, and four nurses) were recruited to fill out the questionnaire and then complete a survey to rank each statement’s clarity and relevance on a scale from 1 to 4 as follows: 1, unclear/irrelevant; 2, somewhat clear/somewhat relevant; 3, clear/relevant; 4, very clear/very relevant.

Finally, the authors requested that the 26 participants (11 men and 15 women; mean age 70.2 ± 7.3 years) fill out the pre-final version and then interviewed them about each statement’s clarity and relevance. In general, no significant difficulties were noted by any respondent, and they were able to read and understand all items. Finally, the Arabic FRQ was produced and ready for psychometric testing.

### 2.2. Psychometric Testing

A licensed physical therapist (the examiner) with 10 years of experience collected data in two sessions. In session 1 (baseline assessment), the therapist began with collecting demographic information including age, sex, height, weight, chronic disease (diabetes and hypertension), marital status, and education level, followed by asking the participants, “Did you fall in the last 12 months?” Participants who were literate were requested to complete the (self-administered) Arabic version of the FRQ, while the questionnaire was administered by the interviewer for those who were nonliterate. To minimize bias in participant responses, the examiner received training on the interviewer-administration method before collecting data for the study. Then, participants participated in three measures: Berg Balance Scale (BBS), Time Up and Go (TUG), and Five Time Sit to Stand (5TSTS; see Section 2.3).

Session 2 (follow-up assessment) occurred seven days later. Participants completed the Arabic version of the FRQ in addition to a self-reported, seven-level global rating of change (GRC) scale to identify any changes in their condition related to the risk of fall compared with the baseline. The GRC included seven choices: (1) completely gone; (2) much better; (3) better; (4) a little better; (5) about the same; (6) a little worse; and (7) much worse. Participants who answered with “about the same”, “a little better”, or “a little worse” were recognized as stable.

The measures administered to the participants are described below.

### 2.3. Measures

#### 2.3.1. Self-Rated FRQ

The FRQ is a valid and reliable scale containing 12 yes/no questions assessing different fall-risk factors in older adults. In this study, participants were instructed to complete the Arabic version of the FRQ. The answers were scored as (No = 0) or (Yes = 2) for the first two questions and (Yes = 1) for the remaining 10 questions. Individuals scoring four or more were considered at increased risk of falling [13].

#### 2.3.2. Berg Balance Scale (BBS)

The BBS is a valid and reliable fall-risk assessment tool for older adults [17]. It consists of 14 functional performance tasks evaluating three aspects: (1) postural adjustment to voluntary movement, (2) reaction to external disturbances, and (3) maintenance of a position. Each task is graded on a 5-point ordinal scale ranging from 0 to 4, with 0 indicating the lowest performance and 4 representing the highest performance level. The total score is 56 points, with higher scores indicating greater independence and a better ability to balance; the cut-off value for assessing the risk of falling varies from 33 to 54 points [18]. In this study, the examiner demonstrated each task for the participants before evaluating their performance.

#### 2.3.3. TUG

TUG is a valid and reliable performance-based fall-risk assessment tool for older adults [19]. It measures the time (in seconds) required to stand up from a standard armchair (height: 45 cm), walk three meters at a comfortable and safe speed, turn, walk back, and sit on the chair again [19]. In this study, the examiner explained and demonstrated the TUG procedure to the participants, after which they performed the test. They were allowed to use their arms for support when standing up or sitting down; the time was recorded using a digital stopwatch. A cut-off score of 12 was used for detecting older adults at high risk for falling [20].

#### 2.3.4. 5TSTS

The 5TSTS is a valid and reliable performance-based fall-risk assessment tool. It measures the time required to transfer from a seated to a standing position and back to sitting five times [21]. In this study, the examiner demonstrated the test to the participants. Then, participants were asked to sit upright on an armless chair with a seat height of 43 cm with their arms crossed on their chest [21]. Then, they performed the test by standing upright (without pushing themselves up) and sitting back down five times without physical assistance and as quickly as possible [22]. The timer was started when the examiner said “go” and was stopped when the participants’ bottom touched the seat following the fifth stand up-sit down cycle [22].

## 3. Data Analyses

The assessment of the Arabic FRQ’s psychometric properties included the evaluation of the questionnaire for floor and ceiling effects, validity, and reliability. A priori hypotheses were predefined, against which the calculated psychometric values were tested (Table 1). The hypotheses were constructed based on the results of previous validation studies [14,15] on the FRQ and the psychometric values recommended by published guidelines. IBM SPSS Statistics for Windows version 25.0 (Armonk, NY, USA) was used to conduct the statistical tests with an α level of 0.05.

### 3.1. Validity

#### 3.1.1. Content Validity

The content validity of the Arabic FRQ was assessed by calculating the item-level content validity index (I-CVI), average scale-level content validity index (S-CVI/Ave), and universal agreement (S-CVI/UA). The I-CVI was calculated by dividing the number of experts who rated an item with a score of 3 or 4 among the total number of experts, the S-CVI/Ave was calculated by summing up the I-CVIs of all items and dividing them by the total number of items, and the S-CVI/UA was calculated by adding all I-CVIs equal to 1.00 divided by the total number of items [25]. It was reported that a value of S-CVI/Ave more than or equal to 0.90 is an indication of excellent content validity of the instrument being tested [24].

#### 3.1.2. Construct (Convergent) Validity

The Arabic FRQ was correlated with the BBS, TUG, and 5TSTS at the baseline. The correlation values were estimated by calculating the Pearson’s correlation coefficient (Pearson’s r). Pearson’s r value was interpreted as follows: <0.25, little or no relationship; 0.25–0.50, fair; 0.50–0.75, moderate; and ≥0.75, excellent [26].

#### 3.1.3. Construct (Known Groups) Validity

The participants were divided into two groups based on their history of falls (fallers and non-fallers). The validity of the Arabic FRQ in identifying those with a history of falls was tested by creating a receiver operating characteristic (ROC) curve to compute the area under the curve (AUC). The AUC ranged between 0.5 (no discriminating accuracy) and 1 (perfect discriminating accuracy) [27]. Subsequently, the optimal cut-off score associated with the highest sensitivity and specificity was identified. The score was determined by locating the point at the uppermost left-hand corner of the ROC curve. Table 1 presents the a priori hypotheses to evaluate the construct validity of the Arabic FRQ. At least 75% of the predefined hypotheses should be accepted to confirm the construct validity of the instrument being evaluated [23].

### 3.2. Floor and Ceiling Effects

The Arabic FRQ was evaluated for the floor and ceiling effects by calculating the number of participants who obtained the lowest (14) or highest (0) status score, respectively. If more than 15% of the participants achieved the maximum or minimum obtainable score, the floor and/or ceiling effects were considered as present [23].

### 3.3. Reliability

Cronbach’s α was calculated to estimate the internal consistency of the Arabic FRQ at baseline. The test-retest reliability between the baseline and seven days later was examined by calculating the intraclass correlation coefficients (ICC) in a two-way random effects model with single measures and absolute agreement. The values of the Cronbach’s α and ICC were interpreted as follows: <0.50, poor; 0.50–0.75, moderate; 0.75–0.90, good; and >0.90, excellent [28,29]. Moreover, the standard error of measurement (SEM) and minimal detectable change at 95% confidence level (MDC_95%_) were calculated using the following formulas: $SEM=SD\sqrt{1-ICC}$ (SD: standard deviation) [26]; $MDC_{95\%}=1.96\;\times \;\sqrt{2}\;\times \;SEM$ [29]. The 95% limits of agreement (LOA) between scores of the two administrations of the Arabic FRQ were assessed by creating a Bland–Altman plot [30]. Only the scores of those participants who were classified as stable in the retest were used to assess reliability.

## 4. Results

The data of 110 participants were used to evaluate the psychometric characteristics of the Arabic FRQ. A total of 36 participants were excluded from the construct validity evaluation of the Arabic FRQ; three could not complete both the TUG and 5TSTS, and thirty-three could not complete the 5TSTS. A total of 40 participants did not attend the follow-up assessment. Out of the remaining 70 participants, 63 were categorized as stable on the GRC and were included in the reliability analyses. Participants’ characteristics, test and retest values for the Arabic FRQ, and values of BBS, TUG, and 5TSTS at baseline are shown in Table 2.

### 4.1. Floor and Ceiling Effects

The Arabic FRQ did not show floor or ceiling effects. At baseline, with 110 participants, two respondents achieved the lowest status score of 14, and three respondents obtained the highest status score of 0. At the follow-up assessment, with 70 participants, only one participant obtained a score of 14, and another participant achieved a score of 0.

### 4.2. Validity

The results of the content validity assessment are summarized in Table 3. For the clarity component, all 12 items were rated 3 or 4 by all 12 experts. All items were considered clear, as none had less than the suggested I-CVI of 0.78. The S-CVI/Ave for clarity was 0.99, signifying an excellent content validity. The S-CVI/UA for clarity was 0.92. For the relevance component, all items had an I-CVI of more than 0.78, except for item 12 (I-CVI = 0.75). However, the S-CVI/Ave for relevance was 0.96, suggesting an excellent level of content validity. The S-CVI/UA was 0.75. All values calculated are consistent with our hypotheses about the content validity of the Arabic FRQ, except for the I-CVI of item 12 and S-CVI/UA for relevance, which were just below the recommended values [24].

### 4.3. Construct Validity

As shown in Table 4, Pearson’s r showed that the Arabic FRQ had a significant moderate negative relationship with BBS as well as fair-to-moderate positive correlations with 5TSTS and TUG, respectively (Table 4).

After constructing the ROC curve, a significant AUC equals to 0.81 (standard error 0.04; 95% CI, 0.73–0.89) was found (Figure 1). The cut-off score was 7.5 and associated with 73.7% sensitivity and 73.6% specificity (Table 4). These values support our hypotheses regarding the construct validity of the translated questionnaire.

### 4.4. Reliability

Table 4 summarizes the reliability values of the Arabic FRQ. The internal consistency was estimated to be good, with a Cronbach’s α of 0.77. Deletion of item 1 slightly increased the Cronbach’s α to 0.78 (Table 5). These data of the Arabic FRQ agree with our reliability hypotheses stated in Table 1.

With 63 participants, the Arabic FRQ demonstrated excellent test-retest reliability with an ICC value of 0.95 (95% CI: 0.92–0.97). The SEM and MDC_95%_ were 0.77 and 2.15, respectively. Moreover, the Arabic FRQ displayed a good distribution of scores with no systematic bias, as shown in the Bland–Altman plot (Figure 2). The mean difference and 95% LOA were equal to 0.27 (−1.71–2.25).

## 5. Discussion

The FRQ is used worldwide as a tool for fall-risk screening in older populations. However, determining this self-rated questionnaire’s cross-cultural validity and reliability is necessary when using it in different contexts. The present study successfully translated, cross-culturally adapted, and psychometrically analyzed the Arabic version of the self-rated FRQ according to international recommendations [16].

The FRQ translation and adaptation from the original English version into Arabic were performed by applying a systematic approach of forward translation, version synthesis, backtranslation, and consolidation of translations with content experts [16]. During the process of translation, cross-cultural adaptation, and pilot testing, no significant modifications of items or responses were needed, and the final version of the Arabic FRQ showed good understanding. Moreover, the psychometric properties of this version were tested among older adults in Saudi Arabia.

The results of this study demonstrate that the Arabic FRQ has fair-to-excellent validity and good-to-excellent reliability in predicting the risk of falls in older adults aged 65 years and older. In particular, the validity of the Arabic FRQ was established in three ways.

First, its content validity was assessed during the transcultural adaptation process by a committee of 12 multidisciplinary experts. Based on their opinion, the content validity of the Arabic FRQ was found to be excellent. To quantify the content validity of the Arabic FRQ, the CVI was computed following the guidelines recommended by Polit et al. [24]. As hypothesized, the Arabic FRQ exhibited excellent content validity, as shown by the CVI values (Table 3). Moreover, the 26 older adults who completed the pre-final Arabic version of the FRQ faced no significant difficulties in answering the questionnaire. They were able to read, comprehend, and complete all 12 items in approximately five minutes. It should be noted that the expert committee translated and adapted the Arabic FRQ using the modern standard Arabic language to enhance the application of the newly adopted Arabic version across other Arabic-speaking countries [31].

Second, the construct convergent validity of the adapted new version was assessed by performing correlation analyses with the fall-related performance-based tests, including BBS, 5TSTS, and TUG. As hypothesized, the Arabic FRQ exhibited a moderate, negative correlation with the BBS (r = −0.72) and a fair-to-moderate, positive correlation with 5TSTS (r = 0.46) and TUG (r = 0.59). The present study’s findings support the construct validity of the adapted Arabic questionnaire. Similar findings were reported for the Thai-FRQ, for which moderate correlations were reported with the BBS (r = −0.69) and TUG (r = 0.53) in patients with osteoporosis [14]. The Turkish FRQ also revealed a fair correlation with the TUG (r = 0.45) when total scores of both test and retest results were analyzed [15]. Further, the Turkish FRQ showed an excellent association with the BBS (r = −0.76) [15]. This correlation is slightly higher than the values obtained in our study and a previous study that translated and adapted the FRQ in the Thai language [14]. This slight variation in the degree of correlation may be due to the disparities between the studies in terms of the populations studied. In the present study, the participants were older adults from the community in Saudi Arabia, while Kitcharanant et al. [14] recruited older adult patients with osteoporosis from an outpatient orthopedic unit. Moreover, the disparity in the degree of correlation between the FRQ and BBS, 5TSTS, and TUG may also be attributable to the notion that these performance-based tests measured different constructs related to balance and risk of falls. This could have further impacted the differences in results between the studies.

Lastly, construct validity of the Arabic FRQ was tested by analyzing the AUC. An AUC value of 0.81 was attained after constructing the ROC curve (Figure 1). This indicates the ability of the Arabic FRQ to distinguish between fallers and non-fallers based on the participants’ reported history of falls. The cut-off value of 7.5 obtained for the Arabic FRQ is greater than the one reported for the original English version of 4 [13]. A possible explanation is the lower sensitivity and specificity associated with the cut-off score in the study. The Arabic FRQ showed 73.7% sensitivity and 73.6% specificity, which are less than the values of the original English version of the FRQ—100% sensitivity and 83.3% specificity. Furthermore, the cut-off score of the original version was associated with a higher sensitivity value, which was also the case for the Arabic FRQ. In fact, a careful examination of the cut-off scores produced for the Arabic FRQ showed that as the score decreases, the sensitivity increases. For example, a cut-off score of 4.5, which is comparable to the one reported for the original FRQ, was associated with 94.7% sensitivity, but the specificity was 45.4%. For this reason, 7.5 was selected, as it was associated with the highest sensitivity and specificity in the ROC curve. Another possible explanation is that 65% of the participants in the validation study of the English FRQ were 80 years of age and older, unlike the present study, which only had 15.5%. This might also explain the higher cut-off score obtained for the Arabic FRQ. Nevertheless, difference in the cut-off values between the English and Arabic FRQ was expected due to cultural differences between Western and Arab societies. For this reason, defining the psychometric properties of a transculturally adapted and self-administered questionnaire is highly recommended to assure optimal utilization of the adapted index [16]. All things considered, the construct validity of the Arabic FRQ is satisfactory; this is supported by confirming all four a priori hypotheses presented in Table 1.

As hypothesized, the study results indicated no floor or ceiling effects in the Arabic FRQ at baseline and at the one-week follow-up assessment. Similar results were found for the Thai FRQ questionnaire for older adults with osteoporosis [14]. This implies that the Arabic FRQ has good content validity, adequate reliability, and a good distribution of scores [23].

The reliability of the Arabic FRQ was examined through evaluation of the internal consistency (Cronbach’s α), test-retest reliability (ICC), SEM, MDC_95%_, and LOA. Internal consistency of the Arabic FRQ was satisfactory with a Cronbach’s α of 0.77, which falls within the recommended values suggested by Terwee et al. [23] (Table 1). The Cronbach’s alpha of 0.77 implies that the 12 items of the Arabic FRQ have good homogeneity, with no lack of association nor redundancy in the items [23]. In comparison with previous research, the Arabic FRQ’s Cronbach’s α was slightly higher compared to the original version developed by Rubenstein et al. [13], which was α = 0.74; this is comparable with the Turkish FRQ, which reported good internal consistency, with a Cronbach’s α coefficient ranging from 0.77 to 0.85 for the test and retest results, respectively [15]. Similarly, good internal consistency of the Thai FRAT was reported with α = 0.78. Higher values were reported in the Thai FRQ, with a Cronbach’s alpha of 0.93 [14].

The test-retest reliability of the Arabic FRQ was excellent, with an ICC of 0.95 for the total score (95% CI: 0.92–0.97), which implies stability over time. These results agree with those reported for the Turkish FRQ (ICC = 0.99) [15]. Similarly, when kappa statistics were used, the test-retest reliability of the Thai FRQ was high, with a kappa value of 1. The kappa results of the total score of the original FRQ was 0.875. These values indicate almost perfect agreement [32] and are comparable with the reliability coefficients reported in the present study. Notably, the current study and the Thai FRQ study performed the test and retest reliability measurements after a one-week interval. The Turkish FRQ study invited 100 older adults aged 65 or more to refill the questionnaire after 15–20 days. Although different test-retest intervals, statistics, and samples were used, the test-retest reliability of the self-rated FRQ and its adopted versions demonstrated excellent reliability. It should be noted that when examining the test-retest reliability of a self-report measures, memory-effect could take place if a short time interval (e.g., 1 to 2 days) is used between the two administrations. On the other hand, change of status could potentially occur within an extended period of time (e.g., 2 weeks). In the current study, therefore, a period of 7 days was determined as a middle course between short and longer intervals from baseline to retest.

To date, no previous reports have examined the value of the absolute reliability of the self-rated FRQ. Therefore, we calculated the SEM, MDC_95%_, and LOA of the Arabic FRQ after a one-week interval. The SEM and MDC_95%_ of the Arabic FRQ were 0.77 and 2.15, respectively. The small score of the SEM in our study indicates that the Arabic FRQ has good absolute reliability. The smaller the SEM of a measurement, the better reliability it has [33]. The mean difference and LOA of the Arabic FRQ were equal to 0.27 (−1.71 to 2.25). Moreover, the Bland–Altman plot of the Arabic FRQ showed good distribution of scores with no heteroscedasticity, as presented in Figure 2.

This study is the first to cross-culturally translate, adapt, and examine the validity and reliability of the Arabic FRQ in older adults. However, it has some limitations. Cognitive impairment testing for the participants in this study was not performed. Another possible limitation is the sample size. Although we believe that this might have minimal impact on the generalizability of the results since both validity and reliability cohorts met the suggested number of 50 participants in validation studies [23], a larger sample size is preferred. An additional limitation of the study is that we did not validate the FRQ against polypharmacy and fear of falling data. Future studies are recommended that evaluate the construct validity of the FRQ against such factors to provide a more comprehensive picture about the validity of the questionnaire. In addition, further research is needed to investigate the responsiveness of Arabic FRQ, the feasibility of the Arabic FRQ in other Arab-speaking communities, and the possibility of using it as an online assessment tool of fall risk.

## 6. Conclusions

The results of this study showed that the self-rated Arabic FRQ is a valid and reliable tool for assessing the risk of falling in older adults aged ≥65 years. This finding would be beneficial in raising awareness, consulting specialists, reducing incidence of falls, and planning fall-prevention programs.

## Figures and Tables

**Figure 1 ijerph-20-05606-f001:**
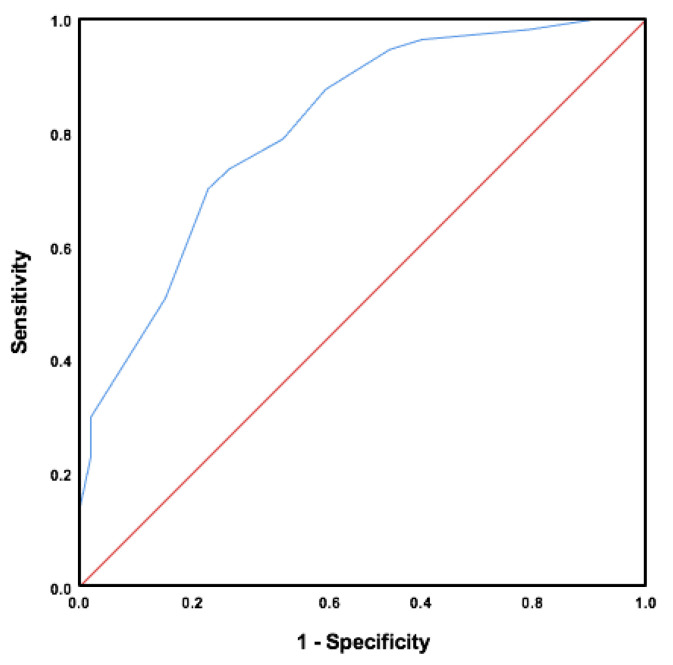
Receiver operating characteristic curve for the Arabic FRQ.

**Figure 2 ijerph-20-05606-f002:**
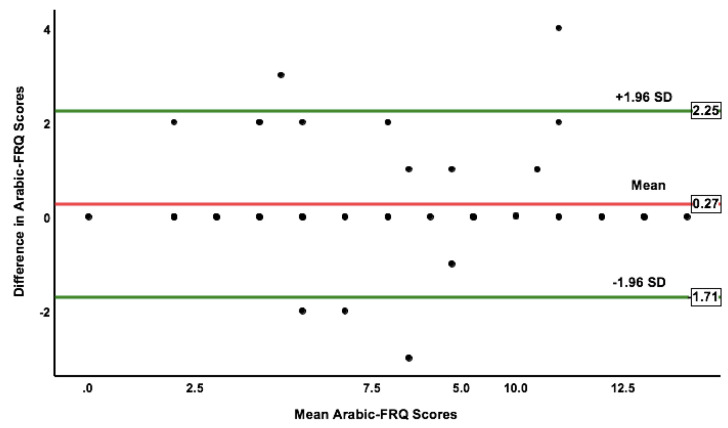
The 95% limits of agreement of the Arabic FRQ scores between the baseline and seven days after. FRQ, Fall Risk Questionnaire; SD, standard deviation.

**Table 1 ijerph-20-05606-t001:** A priori hypotheses for testing the psychometric properties of the Arabic FRQ.

Floor and Ceiling Effects		
**Floor effect**	>15% of the participants achieve the lowest status score [23]	
**Ceiling effect**	>15% of the participants achieve the highest status score [23]	
**Validity**		
**Content validity**	I-CVI ≥ 0.78 [24] S-CVI/Ave ≥ 0.90 [24] S-CVI/UA ≥ 0.80 [24]	
**Instrument/test**	**Construct measured**	**Direction and magnitude of the relationships**
**STS**	Lower-extremity functional strength	Fair-to-moderate positive relationship (r = 0.39–0.65) [14]
**TUG**	Mobility	Fair-to-moderate positive relationship (r = 0.30–0.63) [14,15]
**BBS**	Balance	Moderate-to-excellent negative relationship (r = −0.66 to −0.76) [14,15]
**AUC**	>0.70 [23]	
**Reliability**		
**Internal consistency**	Cronbach’s α = 0.70–0.95 [23]	
**Test-retest**	ICC > 0.70 [23]	

I-CVI, item-level content validity index; S-CVI/Ave, average scale-level content validity index; S-CVI/UA, universal agreement; STS, Five Time Sit to Stand Test; r, Pearson’s correlation coefficient; TUG, Timed Up and Go test; BBS, Berg Balance Scale; AUC, area under the curve; ICC, intraclass correlation coefficient.

**Table 2 ijerph-20-05606-t002:** Participant scores at baseline and follow-up assessment with demographic characteristics.

Tests Scores								
Cohort	Validity Cohort						Reliability Cohort	
**Sample Size**	**n = 110**		**n = 107**		**n = 74**		**n = 63**	
**Test/Statistics**	**M ± SD**	**Min–Max**	**M ± SD**	**Min–Max**	**M ± SD**	**Min–Max**	**M ± SD**	**Min–Max**
**FRQ (baseline)**	7.3 ± 3.6	0–14	7.2 ± 3.6	0–14	6.0 ± 3.2	0–13	7.2 ± 3.6	-
**BBS**	37.5 ± 11.8	14–55	-	-	-	-	-	-
**TUG**	-	-	21.6 ± 8.6	10–50	-	-	-	-
**STS**	-	-	-	-	26.3 ± 7.5	13–54	-	-
**FRQ (retest)**	-	-	-	-	-	-	7.4 ± 3.6	0–14
**Demographic Characteristics ^a^**								
**Age:** **65–69** **70–74** **75–79** **≥80**	49 (44.5) 28 (25.5) 16 (14.5) 17 (15.5)		49 (45.8) 28 (26.2) 14 (13.1) 16 (15)		43 (58.1) 23 (31.1) 5 (6.8) 3 (4.1)		31 (49.2) 16 (25.4) 6 (9.5) 10 (15.9)	
**Sex:** **Male** **Female**	72 (65.5) 38 (34.5)		70 (65.4) 37 (34.6)		47 (63.5) 27 (36.5)		41 (65.1) 22 (34.9)	
**Marital status:** **Married** **Divorced** **Widow**	89 (80.9) 1 (0.9) 20 (18.2)		87 (81.3) 1 (0.9) 19 (17.8)		65 (87.8) 1 (1.4) 8 (10.8)		53 (84.1) 1 (1.6) 9 (14.3)	
**Education:** **Illiterate** **Primary** **Middle** **High** **University**	28 (25.5) 11 (10) 14 (12.7) 20 (18.2) 37 (33.6)		27 (25.2) 11 (10.3) 13 (12.1) 20 (18.7) 36 (33.6)		11 (14.9) 9 (12.2) 10 (13.5) 15 (20.3) 29 (39.2)		19 (30.2) 7 (11.1) 6 (9.5) 10 (15.9) 21 (33.3)	
**Chronic disease:** **Yes** **No**	81 (73.6) 29 (26.4)		78 (72.9) 29 (27.1)		48 (64.9) 26 (35.1)		50 (79.4) 13 (20.6)	
**History of fall:** **Yes** **No**	57 (51.8) 53 (48.2)		55 (51.4) 52 (48.6)		32 (43.2) 42 (56.8)		32 (50.8) 31 (49.2)	
**Weight M ± SD (kg)**	73.4 ± 15.8		73.4 ± 16		73.6 ± 12.5		73.6 ± 18.6	
**Height M ± SD (cm)**	164.1 ± 12.7		164 ± 12.8		165 ± 8.2		162.6 ± 14.7	

M, mean; SD, standard deviation; Min, minimum; Max, maximum; BBS, Berg Balance Scale; Five Time Sit to Stand Test; STS, TUG, Timed Up and Go Test. ^a^ Demographic characteristics are given in number (percentage) unless otherwise indicated.

**Table 3 ijerph-20-05606-t003:** Content validity assessment of the Arabic FRQ.

Variable	Clarity Component	Relevance Component
**Number of items with I-CVI ≥ 0.70**	12	12
**Number of items with I-CVI < 0.70**	0	0
**Minimum–maximum I-CVI**	0.92–1.00	0.75–1.00
**S-CVI/Ave**	0.99	0.96
**S-CVI/UA**	0.92	0.75

I-CVI, item-level content validity index; S-CVI/Ave, average scale-level content validity index; S-CVI/UA, universal agreement.

**Table 4 ijerph-20-05606-t004:** Psychometric properties of the Arabic FRQ.

Psychometric Property		Baseline	At 7 Days	n
**Construct validity**	STS	r = 0.46 *	-	74
	TUG	r = 0.59 *	-	107
	BBS	r = −0.72 *	-	110
	AUC	0.81 * (95% CI = 0.73–0.89)	-	
	Cutoff score (sensitivity–specificity)	7.5 (73.7–73.6%)	-	
**Reliability**	Internal consistency	Cronbach’s α = 0.77	-	
	Test-retest	-	ICC = 0.95 ** (95% CI: 0.92–0.97)	63
	SEM	-	0.77	
	MDC_95%_	-	2.15	
	Mean difference (95% LOA)	-	0.27 (−1.71–2.25)	

STS, Five Time Sit to Stand Test; r, Pearson correlation coefficient; TUG, Timed Up and Go Test; BBS, Berg Balance Scale; AUC, area under the curve; CI, confidence interval; ICC, intraclass correlation coefficient; SEM, standard error of measurement; MDC_95%_, minimal detectable change at 95% confidence level; LOA, limits of agreement. * Two-tailed correlation is significant at α = 0.01. ** Significant at α = 0.05.

**Table 5 ijerph-20-05606-t005:** Internal consistency statistic of the Arabic FRQ.

Item	Cronbach’s α if Item Removed
**1**	0.78
**2**	0.74
**3**	0.73
**4**	0.73
**5**	0.74
**6**	0.75
**7**	0.74
**8**	0.77
**9**	0.74
**10**	0.75
**11**	0.76
**12**	0.75

## Data Availability

The datasets used and analyzed during the current study are available from the corresponding author on reasonable request.

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
