# Peer review of "Cross-Cultural Adaptation and Psychometric Properties of the Arabic Version of the Fall Risk Questionnaire"

_ijerph, 2023, doi:10.3390/ijerph20085606_

Round 1
Reviewer 1 Report
This is very important work since falls are an important public health issue for the elderly population. The methodology is sound and generally well-presented.
Here are my main comments about the work:
- In the introduction, instead of the notion of "increasing specialists consultation", I would suggest putting forward the notion that such scale would prompt seniors to consult specialists when there is a need.
- In the methods section, could you describe the general characteristics of the rehab centers so we can understand they services provided and population they served?
- In the methods, could you clarify or give examples of medical conditions that would fit your inclusion criteria? Also, did you exclude people with cognitive impairements?
- Figure 1: First the Figure is helpful to understand Beatin's suggested process. I would advise authors to obtain the copyright permission to use such similar figure. Can authors explain their rational to slightly derive from Beaton's 5 stages name. For example: Stage 5 in Beaton is "Pretesting" which became "Content validity and Pilot testing" in the manuscript.
- In the methods: Please enhance the description of your translators level of language proficiency. Also, please describe the type of expertise for your methodologist who participared in the fourth stage.
- In the methods: A very important point for stage five is that Beaton's method is clear that each subject should be interviewed and probe about their understand if the items. Could you clarify whether you have done these interviews with their 26 participants? If not, I would add this to the discussion of the study's limitations.
- In the methods: How long was the session 1? Was it part of their physical therapy appointment or course of treatment or completely appart?
- In the methods, authors mentioned IBM SPSS for their Cronbach. Which program did you use for the other statistics?
- Content validity: I'm questionning whether it is appropriate to run a content validity analysis to achieve a cross-cultural adaptation of a validated scale. I'm not sure that adding or removing an item should be a goal. I know Beaton is refering to stage 5 providing "some measure of quality in the content validity", but I'm not sure it meant to run such a complete process.
- In results section: Which item helped to increase the Cronbach? It would be important to mention, eventhough keeping this items is necessary. It would also be helpful if authors would present the classic table of Cronbach's analysis.
- In discussion: Please check the reference 15 used on page 10 as I believe it should be 18.
- In discussion: I would like to see a discussion about the relevance to revise the aformentioned cut-off of 4 to 7.5 of the Arabic-FRQ to identify older adults at risk.
- In discussion: I would add a discussion about the risks and trade-off of using different length of time for the test-retest in the context of a self-rated questionnaire.
- In discussion: Other important limitations should be discussed here: small sample size, a lot of participants lost to follow up, and a sample that is unlikely to be representative of the Saudi poprulation.
- In discussion: It would be important to discuss the fact that this study recruited a population for rehab centers and if the goal is to use the FRQ as a public health tool to prevent falls for seniors living in the community, further studies with that specific population should be conducted.

Reviewer 2 Report
The authors have done an excellent job conducting a rigorous study to evaluate the validity and reliability of the Arabic version of the FRQ. This is an excellent contribution to the body of science around fall risk management, and addresses a huge gap in research and clinical tools. The paper does require some minor revisions before publication. Thanks for the great work and the opportunity to review. I would recommend the following:
Background
· You mention over 50% of older adults in Saudi Arabia have reported a fall in the past year. Can you provide a little more context to this statement – why is the rate so high and does this high rate provide even more reason for your study?
· Starting at line 44 the rationale for the need for the FRQ is confusing and difficult to follow, especially the argument around incorporating fall risk assessments into clinical practice. My confusion is the assessments listed are those that PTs would be using routinely to assess risk. Please clarify that you are specifically talking about physician practice and that MDs or any other healthcare providers would benefit from a quick self-assessment screen to quickly and accurately identify risk. Starting at line 49 about the patient perspective – this should be a new paragraph and should provide a little more context.
· Line 56 – You need a transition statement between the development of the FRQ and the fact it has been validated in other countries. Something like, Other countries have recognized the need for screening and intervention and it looks like the FRQ may be a universal tool
· You mention in the intro, abstract, and conclusion that this is needed for research but you don’t discuss that at all in the paper. I would recommend deleting the value as a research tool.
Methods
· Really well done. My only feedback is line 95-96 – how did the authors and the translators reach consensus? Was their a process in place?
· Psychometric testing – line 130 – the Global Rating of Change Scale – this is the most problematic part of the paper and definitely needs more information. Please describe exactly why it was used and what specific changes in condition was it assessing? The paper states changes related to fall risk compared with baseline but it does not say what was being measured? Was it just the individual’s self assessment or were they being asked to rate change on something specific?
Results
· Exclusion is difficult to follow Maybe want to say a total of 36 participants were excluded, 3 could not complete the TUG or STS and 33 could not complete the STS
Discussion
· You may want to speak to the fact that the FRQ assesses a variety of fall risk factors including polypharmacy and fear of falling and the assessments you used are purely physical assessments which may account for the correlation values
· Limitations – the main limitation is that it was a sample of convenience and may not be representative of the broader population. Also the responsiveness is not a limitation – it is a future study – this study was not asking that question.
